# The Nursing Work Environment, Supervisory Support, Nurse Characteristics, and Burnout as Predictors of Intent to Stay among Hospital Nurses in the Republic of Korea: A Path Analysis

**DOI:** 10.3390/healthcare11111653

**Published:** 2023-06-05

**Authors:** Young-Bum Kim, Seung-Hee Lee

**Affiliations:** 1Department of Sociology, Institute of Aging, Hallym University, Chuncheon 24252, Republic of Korea; twoponej@hallym.ac.kr; 2Department of Nursing, The University of Ulsan, Ulsan 44610, Republic of Korea

**Keywords:** work environment, burnout, intent to stay, nurse, Republic of Korea

## Abstract

This study aimed to examine the comprehensive impact of five aspects of the nursing work environment as well as supervisory support, nurse characteristics, and burnout on intent to stay (ITS) among Korean hospital nurses. A cross-sectional questionnaire was distributed in seven general hospitals from May to July of 2019. Data were collected from a sample of 631 Korean nurses. The hypothesized model was evaluated using the STATA program for path models. Findings demonstrated that burnout played a mediating role on the relationships between the nursing work environment, supervisory support, nurse characteristics, and ITS. Burnout was the most influential predictor of ITS (*β* = −0.36, *p* < 0.001). Nurse participation in hospital affairs (*β* = 0.10, *p* = 0.044) and collegial nurse–physician relationships (*β* = 0.08, *p* = 0.038) had a direct effect on ITS. Supervisory support had a significant direct effect on ITS (*β* = 0.19, *p* < 0.001). Therefore, to increase nurses’ ITS, it is necessary to improve their participation in hospital affairs and collegial relationships, as well as strengthen support from supervisors and reduce burnout.

## 1. Introduction

Globally, the shortage and high turnover rate of nurses are significant issues that threaten both patient safety and quality of nursing care [1]. Nurses are essential members of the workforce in the health care system. Retaining nurses who already work in hospitals would be a partial solution to the nursing shortage [2]. The intent to stay (ITS) at work has been known to be the strongest predictor of nurse retention [3]. ITS is nurses’ perception of their likelihood of staying in the current job [2]. A better understanding of nurses’ ITS may lead to improvement in nurse retention and quality of patient care. Policy makers and nurse leaders who aim to design effective nurse retention strategies must understand the variables that are associated with nurses’ ITS.

Studies of nurses in hospitals have shown a positive association between favorable work environments and ITS [4,5]. Nurses working in better work environments have reported higher ITS. The nursing work environment is defined as the organizational characteristics of a work setting that foster or hamper professional nursing practice [6]. Lake [6] suggested that the nursing work environment is composed of five aspects: nurse participation in hospital affairs; nursing foundations for quality care; nurse managers’ ability, leadership, and support; staffing and resource adequacy; and collegial nurse–physician relationships. Although numerous studies have investigated the association between the nursing work environment and ITS, we know little about which of these five aspects contributes the most to nurses’ ITS. To our knowledge, no studies have examined the link between these five aspects of the nursing work environment and ITS among hospital nurses. Knowledge about how various aspects of the nursing work environment relate to ITS is needed to develop optimal policies for enhancing nurses’ ITS.

Supervisory support is defined as the assistance and caring that employee receive from supervisors or managers [7]. In a hospital setting, supportive supervisors can help nurses perform their jobs more effectively and provide work guidelines to improve work processes [8]. Manager characteristics such as supervisory support, praise, recognition, and leadership have also been known to affect nurses’ ITS [1,9]. Nurses showed an increased ITS when perceiving strong supervisory support [10]. Nurses’ personal characteristics, such as age, experience as a nurse, marriage, education, and position could also influence ITS [9,11,12]. Older nurses were more likely to remain employed than younger nurses [11]. The intent to stay increased as the nurses’ years of experience increased [12]. Married, and less educated nurses are more likely to remain in their current positions than their counterparts [1,9].

Nurse burnout has been found to be negatively related to nurses’ ITS [9]. Additionally, some studies among hospital nurses have suggested that burnout might be an important mediator in the links between the nursing work environment and some job outcome variables such as turnover intention and ITS [13,14]. Nurses who perceived their work environment as negative experienced more burnout than those who did not [15]. Burnout reduces nurses’ ITS [14], increases nurses’ turnover intention [13]. Burnout resulting from an unfavorable work environment increased nurses’ turnover intention [13], reduced nurses’ ITS [14]. Moreover, a study has shown that nurses’ greater participation in hospital affairs reduces their likelihood of burnout [16]. Another study has found that better relationships between nurses and physicians reduce a likelihood of nurses’ burnout [14]. Other study examining the association between five aspects of the nursing work environment and burnout showed that nurse burnout was significantly linked to nursing foundations for quality of care, nurse participation in hospital affairs, and adequate staffing [17]. Research has also found that when nurses are provided with sufficient support from supervisors and organizational support, they tend to experience lower levels of burnout [18,19]. Reduced burnout with adequate supervisory support may lead to improved ITS. Although many studies have investigated the factors affecting nurses’ ITS, we know little about the mechanisms underlying the relationships among five aspects of the nursing work environment, supervisory support, burnout, and ITS. A theoretical model can broaden our understanding of the structural relationships between ITS and its associated factors. Cowden and Cummings [9] presented a theoretical model of hospital nurses’ ITS. The model included several variables that directly or indirectly affect nurses’ ITS: work characteristics, organization characteristics, manager characteristics, and nurse characteristics. Moreover, in the model, nurses’ affective responses to work (e.g., burnout, job satisfaction) have a mediating position between work characteristics, organization characteristics, manager characteristics, and nurse characteristics, and ITS [9]. ITS is affected by nurses’ affective responses to their work environment such as burnout. Cowden and Cummings have proposed that affective responses (e.g., burnout) are not only the important predictors of ITS but also play mediating roles in ITS [9,20].

Based on Cowden and Cummings’s model and other literature, we used a conceptual model to guide the present study (Figure 1). The model of this study proposes that nurses’ ITS is directly influenced by the nursing work environment (work characteristics and organization characteristics), supervisory support (manager characteristics), and nurse characteristics and is indirectly influenced by these variables through burnout (affective responses). Burnout will mediate the relationships among the nursing work environment, supervisory support, nurse characteristics, and ITS. In particular, we analyzed which of the five aspects of the nursing work environment contributed the most to nurses’ ITS. The current study attempts to reveal the potential mechanisms between five aspects of the nursing work environment, supervisory support, nurse characteristics, and burnout, and ITS among hospital nurses. The nursing work environment is highly associated with nurses’ ITS, and more empirical evidence are needed to investigate the mechanism underlying these relationships among nurses. Our study focuses on the mediating role of burnout in the links among variables. This study aimed to examine the comprehensive impact of five aspects of the nursing work environment, supervisory support, nurse characteristics, and burnout on ITS among Korean hospital nurses.

## 2. Materials and Methods

### 2.1. Study Design

This study used a cross-sectional, correlational design and proposed a hypothetical model for exploring the comprehensive impact of five aspects of the nursing work environment, supervisory support, nurse characteristics, and burnout on ITS in Korean hospital nurses.

### 2.2. Participants and Data Collection

The current study was carried out with nurses in seven general hospitals in Ulsan, an industrial city in the Republic of Korea, who were recruited using convenience sampling. Eligible participants were registered nurses working in either medical, surgical, or intensive care units, with at least one year of clinical experience, and who agreed to participate in this study. The hospitals were general acute care hospitals with 205 to 569 beds. Data were collected from May to July of 2019. A researcher visited each hospital and explained the aim and method of this study to nursing staff on each nursing unit. Nurses willing to participate in this study were given an envelope that included a cover letter explaining the study’s aim and procedures along with a questionnaire. A total of 750 eligible nurses of seven hospitals were invited to participate in this study, with a final sample of 631 participants (84.1% response rate). For a path analysis, the required sample size should include 15 times each measured variable [21]. Hence, the sample size of 631 was sufficient for testing a hypothesized model with 12 measured variables.

### 2.3. Instruments

#### 2.3.1. Nurse Characteristics

The following four nurse characteristics were entered into the path model as continuous or categorical variables: age, marital status, education, and position. The characteristics of nurses were collected via single-item survey questions as individual variables for each characteristic.

#### 2.3.2. The Nursing Work Environment

The Korean version of the Practice Environment Scale of the Nursing Work Index (K-PES-NWI), initially developed by Lake [6] and validated by Cho and colleagues [22], was used to measure the nursing work environment. The K-PES-NWI assesses various aspects of nurses’ work environments and consists of 29 items categorized into five dimensions: (1) nurse participation in hospital affairs (nine items); (2) nursing foundations for quality care (nine items); (3) nurse managers’ ability and leadership and support of nurses (four items); (4) staffing and resource adequacy (four items); and (5) collegial nurse–physician relationships (three items). Each item was measured using a four-point Likert scale (1 = strongly disagree, 2 = disagree, 3 = agree, 4 = strongly agree). The mean subscale scores were calculated by averaging item scores in each subscale. A higher score indicated more favorable perception of the nursing work environment. Cronbach’s alphas for the PES-NWI subscales were reported to range from 0.71 to 0.84 in Lake’s study [6]. Cronbach’s alphas for the K-PES-NWI subscales ranged from 0.80 to 0.84 in Cho et al.’s study [22]. In the present study, the overall alpha coefficient for the scale was 0.90, and the subscale coefficients ranged from 0.69 to 0.80. These internal consistency reliabilities were acceptable, but the nurse manager ability, leadership, and support of nurse subscale was lower than reported previously [6,22].

#### 2.3.3. Supervisory Support

Supervisory support was measured using the instrument originally developed by LaRocco et al. [23] and translated by Jung [24]. This scale comprises eight items (e.g., “My immediate manager listens well to issues related to work performance”; “I can turn to my immediate manager when I get into a difficult problem at work”) arranged on a five-point Likert-type scale (1 = strongly disagree, 5 = strongly agree). A higher score indicates a stronger feeling of supervisory support. The Cronbach’s α of the tool was 0.93 in Jung’s study [24] among Koreans. In this study, it was 0.92.

#### 2.3.4. Burnout

Burnout was measured using the nine-item emotional exhaustion (EE) subscale of the Maslach Burnout Inventory (MBI) [25]. The EE subscale of the MBI, which describes the core element of burnout, has been used widely in nursing studies as a measure of burnout [26]. It has been reported to have good validity and reliability [26]. This study used the Korean version of the EE subscale translated by Kang and Kim [27]. The Cronbach’s α was 0.91 in the Republic of Korea [27] and 0.90 in the current study. Each item was measured using a seven-point Likert scale (0 = never, 6 = every day); the higher score, the higher the participant’s level of burnout.

#### 2.3.5. Intent to Stay (ITS)

ITS was measured using the nurse retention index (NRI), developed by Cowin [28]. The NRI assesses nurses’ ITS in their current nursing job, and it consists of six items arranged on an eight-point Likert-type scale (1 = strongly disagree, 8 = strongly agree). The tool has four positively worded items and two negatively worded items (reverse coded). A higher score indicates higher ITS. The validity and reliability of the NRI have been well established by existing research, with acceptable psychometric properties [29]. The present study used the Korean version of the NRI translated by Kim [30]. Cronbach’s α of the tool was 0.89 among Korean nurses and 0.91 in this study.

### 2.4. Ethical Considerations

Prior to data collection, ethical approval (IRB no. 1040968-A-2019-003) was obtained from the Institutional Review Board where the author is affiliated. Participants were informed that data confidentiality and respondent anonymity would be assured, that they could withdraw from this study whenever they wanted without disadvantages, and that the data would be kept in a locked file cabinet. Participants provided written informed consent, voluntarily completed the questionnaire, and returned it in a concealed envelope.

### 2.5. Statistical Analysis

STATA, version 15.1 was used for data analysis. A statistical significance level was set at *p* < 0.05 (two-tailed). Descriptive statistics were calculated for all study variables. Correlations between the variables were estimated using Pearson’s correlation coefficients. Scale reliability was assessed by computing Cronbach’s alpha coefficient, as a measure of internal consistency. The hypothesized model was then evaluated using the STATA software program for path models, employing a maximum likelihood estimation. Path analysis allows for simultaneous testing of both the direct and indirect effects of independent variables on dependent variables. In this analysis, the following fit indices were used to evaluate the validity and fitness of the model: non-significant χ^2^ value, degrees of freedom (df), the Tucker–Lewis index (TLI), the comparative fit index (CFI), root mean square error of approximation (RMSEA), and standardized root mean square residual (SRMR). A model is evaluated to have a good fit if the CFI and TLI are 0.95 or greater, the RMSEA value is lower than 0.08 and the SRMR value is <0.08 [31]. All study variables in our model have been rigorously tested with diverse study populations as well as in this study regarding validity and reliability [6,22,23,24,25,26,27,28,29,30]. In addition, we computed the squared multiple correlations (SMC) of the study variables on burnout and ITS in the path model.

## 3. Results

### 3.1. Participant Characteristics

The 631 participants (Table 1) had a mean age of 31.23 (SD = 6.37) years (range: 22–53) and had, on average, 7.38 years of clinical nursing experience (SD = 6.26). Most participants were women (*n* = 624; 98.9%). Approximately 68% of the participants were unmarried and 91% were staff nurses. The education levels of participants included Bachelor’s degrees and above (*n* = 393; 62%) and college diplomas (*n* = 238; 37.7%).

### 3.2. Descriptive Statistics and Correlations

Table 2 presents the means, standard deviations, Cronbach’s alphas, and correlation coefficients for the key variables. The Cronbach’s alphas for the study variables were between 0.69 and 0.92. As anticipated, correlation analyses showed that ITS was positively related to nurse participation in hospital affairs (*r* = 0.29, *p* < 0.001); nurse foundation for quality care (*r* = 0.24, *p* < 0.001); nurse manager ability; leadership and support of nurses (*r* = 0.30, *p* < 0.001); staffing and resource adequacy (*r* = 0.27, *p* < 0.001); collegial nurse–physician relationships (*r* = 0.30, *p* < 0.001); and supervisory support (r = 0.37, *p* < 0.001). Additionally, a significant negative relationship between burnout and ITS was observed (*r* = −0.51, *p* < 0.001).

### 3.3. Evaluation of the Path Model

To examine our hypothesized path model validity, we used the non-significant χ^2^ value, df, TLI, CFI, RMSEA, and SRMR. The fit indices of final model showed favorable convergent validity with the TLI = 1.000, CFI = 1.000, RMSEA = 0.000 and SRMR = 0.000. Although the chi-square value was significant (χ^2^ = 499.860, df = 21, *p* < 0.001), it is not appropriate for the fit index when the sample size is large. The reliability of the study variables was verified via Cronbach’s alpha coefficients. As shown in Table 2, Cronbach’s alpha values for variables were between 0.69 and 0.92. Thus, the following hypothetical path model was chosen as the final model for this study. When examining all proposed paths, 9 of 15 paths were significant (Figure 2).

### 3.4. Analysis of the Variable Effects

The direct, indirect, and total effects of exogenous variables on endogenous variables in this path model are presented in Table 3. Nurse participation in hospital affairs had significant direct and total effects, respectively, on ITS (*β* = 0.10, *p* = 0.044; *β* = 0.14, *p* = 0.045). Burnout had significant direct and total effects, respectively, on ITS (*β* = −0.36, *p* < 0.001; *β* = −0.36, *p* < 0.001). Collegial nurse–physician relationships had a significant direct effect on burnout (*β* = −0.16, *p* < 0.001) and ITS (*β* = 0.08, *p* = 0.038) and an indirect effect on ITS (*β* = 0.06, *p* < 0.001). This finding indicates that burnout mediates the path of collegial nurse–physician relationships to ITS. Staffing and resource adequacy had a significant direct effect on burnout (*β* = −0.28, *p* < 0.001) and an indirect effect on ITS (*β* = 0.10, *p* < 0.001). Staffing and resource adequacy had no significant direct effect on ITS (*β* = −0.03, *p* = 0.489). This means that burnout mediates the path of staffing and resource adequacy to ITS.

Supervisory support had a significant direct effect on burnout (*β* = −0.15, *p* < 0.001) and ITS (*β* = 0.19, *p* < 0.001) and an indirect effect on ITS (*β* = 0.05, *p* < 0.001). This result indicates that burnout mediates the path of supervisory support to ITS. In other words, supervisory support is significantly associated with ITS via the effects of burnout. Marriage had a significant direct effect on burnout (*β* = −0.10, *p* = 0.027) and an indirect effect on ITS (*β* = 0.04, *p* = 0.031). Marriage had no significant direct effect on ITS (*β* = 0.07, *p* = 0.106). This indicates that burnout mediates the path of marriage to ITS. Age had significant direct and total effects on ITS (*β* = 0.26, *p* < 0.001; *β* = 0.28, *p* < 0.001). Education also had significant direct and total effects on ITS (*β* = 0.07, *p* = 0.025; *β* = 0.08, *p* = 0.026). Additionally, in order to determine which variable has the greater effect on ITS, a comparative analysis of the standardized total effects (*β*) of the variables on ITS was undertaken. The variables that had the greater impact on ITS were burnout (*β* = −0.36, *p* < 0.001), age (*β* = 0.28, *p* < 0.0001), supervisory support (*β* = 0.24, *p* < 0.001), nurse participation in hospital affairs (*β* = 0.14, *p* = 0.045), and collegial nurse–physician relationships (*β* = 0.14, *p* = 0.039), in that order. Lastly, based on the SMC values, nurse participation in hospital affairs, staffing and resource adequacy, collegial nurse–physician relationships, supervisory support, age, marriage, and education explained the total variance of ITS by 38.5%.

## 4. Discussion

The aim of this study was to examine the comprehensive impact of five aspects of the nursing work environment, supervisory support, nurse characteristics, and burnout on ITS among Korean hospital nurses. Study results from path analyses supported the hypothesized model generally but not all hypothesized paths. Some model paths were not significant.

Two of the five aspects of the nursing work environment had a direct effect on ITS. Nurse participation in hospital affairs directly influenced ITS, supporting our hypothesized model. Nurses were more likely to report higher levels of ITS when they perceive high levels of nurse participation in hospital affairs. Involving nurses in hospital activities and decisions may enhance their autonomy and job engagement, which is a crucial work-related factor that has been linked to ITS [32,33]. Our finding was the same as that in previous studies: that lower nurse participation in hospital affairs was associated with higher intent to leave [34]. Nurse participation in hospital affairs was not significantly linked to burnout, inconsistent with previous studies [17,35]. This finding did not support the proposed hypotheses. Studies have found that nurses’ greater participation in hospital affairs reduced their likelihood of burnout [17,35]. If nurses take more active roles in decision-making for hospital affairs, they might perceive themselves as empowered and decrease their risk of burnout [17]. Further research testing clearly the association between nurse participation in hospital affairs and burnout is needed.

Collegial nurse–physician relationships directly influenced ITS. Nurses who perceived high levels of collegial nurse–physician relationships would have higher levels of ITS. This is resembling findings of previous studies that good relationships with physicians were positively related with ITS [36]. Collegial nurse–physician relationships can improve work commitment and job satisfaction, which impact ITS [36,37]. Collegial nurse–physician relationships were also tied to lower levels of burnout, consistent with existing research [35]. These findings supported our hypotheses that collegial nurse–physician relationships are associated with ITS directly and indirectly, and burnout plays a mediating role on the relationships between collegial nurse–physician relationships and ITS. This was in line with studies that revealed collegial nurse–physician relationships were associated with lower levels of burnout [35], and burnout mediated the relationship between nurse–physician relations and turnover intentions [38].

Staffing and resource adequacy did not directly influence nurses’ ITS, but indirectly influenced ITS through the burnout. This finding partially supports our hypothesized model. Staffing and resource adequacy had a strong negative influence on burnout. Burnout held a mediating position between staffing and resource adequacy and ITS. These results support previous evidence that sufficient staffing and resources could reduce the risk of nurse burnout, thereby promoting ITS [17,35]. Chronic understaffing and resource inadequacy may cause unsustainable workloads, eventually leading to increased burnout [35]. Accordingly, in order to prevent nurses from experiencing burnout, it is crucial to keep staffing aligned with the actual job demands and retain adequate resources. unsustainable

Supervisory support directly influenced ITS. Nurses who perceived high levels of supervisory support would have higher levels of ITS. This is in accord with prior literature reporting the associations between supervisory support and ITS, turnover intention in nurses [9,38]. Supervisory support can make nurses feel that their work is appreciated, to be satisfied with their job, and encourage them to intend to stay at work [11]. Supervisory support also had a negative influence on burnout, in line with prior investigations [39]. Nurses who experience sufficient support from supervisors appear better able to cope with the potential negative effects of work pressures and can experience less burnout [39], more job satisfaction [38], leading to enhance ITS [31]. These findings supported our hypotheses that supervisory support is link to ITS directly and indirectly, and burnout plays a mediating role on the relationships between supervisory support and ITS. Our finding demonstrates the importance of supervisory support in reducing nurse burnout, further enhancing ITS. In the present study, only two nurse characteristics (age and education) were found to have significant direct effects on nurses’ ITS. Older nurses were more likely to stay with their job than younger nurses, consistent with previous findings [1]. Nurses with bachelor’s degrees and above were more likely to stay with their job than nurses with a diploma, which is dissimilar to other findings [25]. Marriage had a negative direct effect on nurse burnout. Married nurses had lower burnout compared with unmarried nurses, inconsistent with existing findings that married health care workers have higher burnout level than their unmarried counterparts because of family roles and work conditions [40].

In the current study, burnout directly and negatively affected ITS, supporting our hypothesized model. When nurses are burned out, they may feel they are not doing their best to care for patients, cannot communicate properly with others, lack motivation, and have poor job-related self-esteem [41]. Eventually, burnout may contribute to nurses leaving their positions [41]. Burnout also played a mediating role on the relationships among staffing and resource adequacy, collegial nurse–physician relationships, supervisory support, marriage, and ITS in this study. In addition, burnout was the most crucial predictor of ITS in the present study. This finding highlights the importance of burnout for enhancing ITS of nurses. Thus, nurse managers should systematically measure burnout and design proper tactics to reduce burnout of nurses. Furthermore, we examined the effect of each aspect of the nursing work environment on ITS. The findings showed that among the five aspects, nurse participation in hospital affairs had direct positive effect on ITS and collegial nurse–physician relationships had direct and indirect positive effect on ITS. To our knowledge, the present study was the first study to explore how five aspects of the nursing work environment, supervisory support, nurse characteristics, and burnout contribute to ITS among hospital nurses. Our findings extend understanding of how five aspects of the nursing work environment, supervisory support, and nurse characteristics are related to ITS through burnout. These results are important, because it provides nurse managers with empirical guidance to help to improve ITS of nurses. In this study, the hypothesized model was tested using path analysis, not structural equation modeling (SEM) for two reasons. First, the psychometrics of each measurement used in this study not only have been well established in existing studies, but have also been validated in the Korean version. Secondly, not including measurement model can reduce complexity given the many study variables and complex pathways in our model. Thus, in order to focus on analyzing direct and indirect effects, which is the main goal of this study, our model was tested via path analysis, not employing SEM.

It is vital to note limitations of this research, and directions for further studies. This study had several limitations. First, we used cross-sectional data, which may have limited causality. Thus, a longitudinal study is warranted to confirm causal relationships between the nursing work environment, supervisory support, nurse characteristics, burnout and ITS. Second, our measures of the nursing work environment were self-reported and could not determine the actual work environment characteristics. Third, since we used convenience sampling in this study, the generalization of the findings might be limited. The revised model resulting from this study needs to be tested in future studies with a more diverse cohort. Further research using a randomized sampling technique with longitudinal designs is warranted to generalize and extend the study findings. Despite these limitations, our study adds important information to the current literature by exploring the comprehensive impact of five aspects of the nursing work environment, supervisory support, nurse characteristics, and burnout on ITS among Korean hospital nurses.

## 5. Conclusions

This study firstly explored the specific causal pathways between the five aspects of the nursing work environment, supervisory support, nurse characteristics, burnout, and ITS. The current study provides nurse managers with some meaningful implications to enhance ITS of nurses. In terms of the nursing work environment, nurse participation in hospital affairs and collegial nurse–physician relationships showed significant direct positive effects on ITS unlike the other aspects of the nursing work environment. In particular, burnout was the most influential predictor of ITS. Burnout played a mediating role on the relationships among staffing and resource adequacy, collegial nurse–physician relationships, supervisory support, marriage, and ITS in this study. In order to increase nurse’ ITS, hospital executives and nurse managers should take corresponding measures to reduce nurse burnout and improve the nursing work environment continuously.

## Figures and Tables

**Figure 1 healthcare-11-01653-f001:**
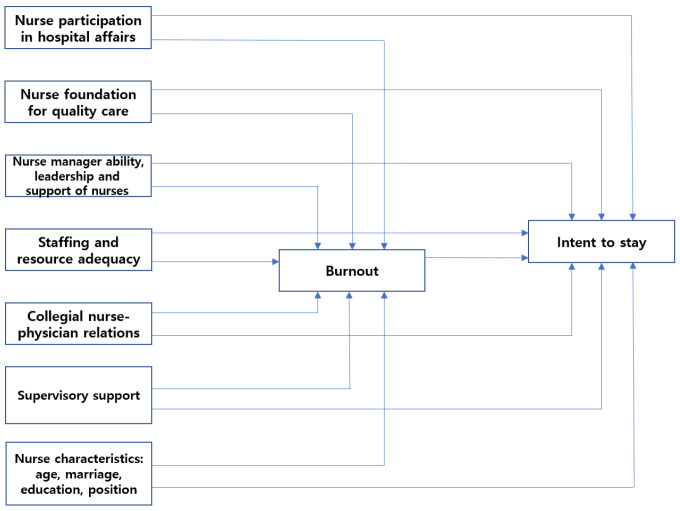
The conceptual model to guide this study.

**Figure 2 healthcare-11-01653-f002:**
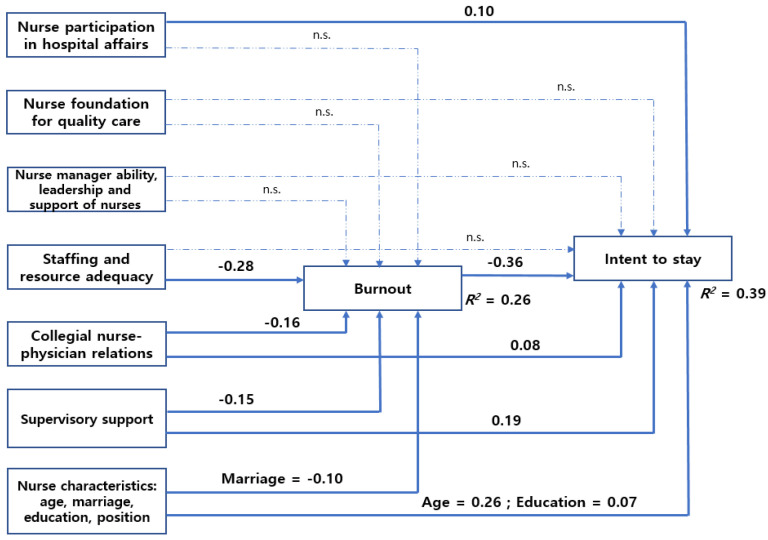
Results of the path analyses of the hypothesized model. n.s., non-significant.

**Table 1 healthcare-11-01653-t001:** Demographic characteristics of participants (*n* = 631).

Variables	*n*	%	Mean	SD	Range
Age (years)			31.23	6.37	22–53
Gender					
Female	624	98.9			
Male	7	1.1			
Educational level					
Diploma	238	37.7			
Bachelor degree and above	393	62.3			
Working experience (years)			7.38	6.26	1–32
Position					
Staff nurse	574	91.0			
Head nurse	57	9.0			
Marital status					
Unmarried	426	67.5			
Married	205	32.5			

Note. *n*, frequency; SD, standard deviation.

**Table 2 healthcare-11-01653-t002:** Descriptive statistics, Cronbach’s alpha coefficients and correlations of study variables.

Variables	Mean	SD	*α*	1	2	3	4	5	6	7	8
1. Intent to stay	4.93	1.47	0.91	_							
2. Burnout	4.22	0.52	0.90	−0.51 ***	_						
3. Nurse participation in hospital affairs	2.50	0.40	0.80	0.29 ***	−0.36 ***	_					
4. Nurse foundation for quality care	2.53	0.35	0.74	0.24 ***	−0.30 ***	0.70 ***	_				
5. Nurse manager ability, leadership and support of nurses	2.78	0.52	0.69	0.30 ***	−0.32 ***	0.58 ***	0.54 ***	_			
6. Staffing and resource adequacy	2.41	0.44	0.71	0.27 ***	−0.42 ***	0.56 ***	0.51 ***	0.48 ***	_		
7. Collegial nurse–physician relations	2.72	0.46	0.79	0.30 ***	−0.33 ***	0.47 ***	0.44 ***	0.46 ***	0.33 ***	_	
8. Supervisory support	3.56	0.67	0.92	0.37 ***	−0.32 ***	0.37 ***	0.40 ***	0.50 ***	0.29 ***	0.35 ***	_

Note. *α*, Cronbach’s alpha coefficients. *** *p* < 0.001.

**Table 3 healthcare-11-01653-t003:** Standardized effects of final model.

Endogenous Variables	Exogenous Variables	Direct Effect		Indirect Effect		Total Effect		SMC
		*β*	*p*	*β*	*p*	*β*	*p*	
Burnout	Nurse participation in hospital affairs	−0.10	0.054			−0.10	0.054	0.263
	Nurse foundation for quality care	0.03	0.516			0.03	0.516	
	Nurse manager ability, leadership and support of nurses	0.01	0.873			0.01	0.873	
	Staffing and resource adequacy	−0.28	<0.001			−0.28	<0.001	
	Collegial nurse–physician relations	−0.16	<0.001			−0.16	<0.001	
	Supervisory support	−0.15	<0.001			−0.15	<0.001	
	Nurse characteristics: age	−0.05	0.349			−0.05	0.349	
	Marital status (married = 1)	−0.10	0.027			−0.10	0.027	
	Education (bachelor and above = 1)	−0.02	0.614			−0.02	0.614	
	Position (head nurse = 1)	0.01	0.777			0.01	0.777	
Intent to stay	Nurse participation in hospital affairs	0.10	0.044	0.04	0.059	0.14	0.045	0.385
	Nurse foundation for quality care	−0.01	0.844	−0.01	0.517	−0.02	0.844	
	Nurse manager ability, leadership and support of nurses	0.01	0.806	−0.00	0.873	0.01	0.806	
	Staffing and resource adequacy	−0.03	0.489	0.10	<0.001	0.07	0.489	
	Collegial nurse–physician relations	0.08	0.038	0.06	<0.001	0.14	0.039	
	Supervisory support	0.19	<0.001	0.05	0.001	0.24	<0.001	
	Nurse characteristics: age	0.26	<0.001	0.02	0.351	0.28	<0.001	
	Marital status (married = 1)	0.07	0.106	0.04	0.031	0.10	0.106	
	Education (bachelor and above = 1)	0.07	0.025	0.01	0.615	0.08	0.026	
	Position (head nurse = 1)	−0.06	0.127	−0.00	0.777	−0.07	0.127	
	Burnout	−0.36	<0.001			−0.36	<0.001	

Note: *β*, standardized parameter estimate; SMC, squared multiple correlation.

## Data Availability

Data sharing is not applicable.

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
