# Peer review of "The Nursing Work Environment, Supervisory Support, Nurse Characteristics, and Burnout as Predictors of Intent to Stay among Hospital Nurses in the Republic of Korea: A Path Analysis"

_healthcare, 2023, doi:10.3390/healthcare11111653_

Round 1

Reviewer 1 Report

Well done!  Understandable presentation of findings, path figure self-explanatory, discussion of findings placed within context of hypothetical model.

Author Response

Thank you for the thoughtful review of the ms. Entitled “Nursing work environment, supervisory support, nurse characteristics, and burnout as predictors of intent to stay among hospital nurses in South Korea: A path analysis”

Reviewer 2 Report

Thank you very much for the opportunity to review an interesting and important article on the subject of Nursing work environment, supervisory support, nurse characteristics, and burnout as predictors of intent to stay among hospital nurses in South Korea: A path analysis

Several suggestions for corrections / additions to be made:

Introduction:

Did the authors have checked if age or seniority were predictors for intention to stay?

In addition, does the place of work have meaning, i.e. the hospital ward where you work with the intention of staying?

Please add to the introduction an update references.

Also, the research rationale should be refined at the end of the introduction while presenting the model.

The conclusions should be adapted to the results of the current study and the innovation in it should be emphasized.

Author Response

Thank you for the thoughtful review of the ms. Entitled “Nursing work environment, supervisory support, nurse characteristics, and burnout as predictors of intent to stay among hospital nurses in South Korea: A path analysis” We have revised the manuscript following the suggestions and recommendations made in the reviews and are resubmitting it for the consideration for publication.

We have responded to the critical points in the review in the following way. The comments are italicized.

Reviewer 2

Thank you very much for the opportunity to review an interesting and important article on the subject of Nursing work environment, supervisory support, nurse characteristics, and burnout as predictors of intent to stay among hospital nurses in South Korea: A path analysis

Several suggestions for corrections / additions to be made:

Point 1: Introduction:

Did the authors have checked if age or seniority were predictors for intention to stay? In addition, does the place of work have meaning, i.e. the hospital ward where you work with the intention of staying?

We have checked if age or seniority were predictors for intention to stay. Age and experience as a nurse were positively associated whit intent to stay of nurses. Older nurses were more likely to remain employed than younger nurses. The intent to stay increased as the nurses’ years of experience increased. However, there have been more studies that suggested age rather than nurses’ years of experience as a predictor for intention to stay. Because of the high correlation between the two variables (age and years of experience), only age was selected for analysis in this study.

Added sentences are as follows.

In the introduction section

Nurses’ personal characteristics, such as age, experience as a nurse, marriage, education, and position could also influence ITS [9,11,12]. Older nurses were more likely to remain employed than younger nurses [11]. The intent to stay increased as the nurses’ years of experience increased [12].

In addition, does the place of work have meaning, i.e. the hospital ward where you work with the intention of staying?

Yes, it does. In this study, the place of work means the place where nurses work such as hospital ward and intensive care unit.

Point 2: Please add to the introduction an update references.

Introduction section has been updated with recent references.

Updated sentences are as follows.

In the introduction section

Globally, the shortage and high turnover rate of nurses are significant issues that threaten both patient safety and quality of nursing care [1]. Nurses are essential members of the workforce in the health care system. Retaining nurses who already work in hospitals would be a partial solution to the nursing shortage [2]. The intent to stay (ITS) at work has been known to be the strongest predictor of nurse retention [3]. ITS is nurses’ perception of their likelihood of staying in the current job [2].

In a hospital setting, supportive supervisor can help nurses do their jobs more effectively and provide work guidelines to improve work processes [8]. Manager characteristics such as supervisory support, praise, recognition, and leadership have also been known to affect nurses’ ITS [1,9]. Nurses showed an increased ITS when perceiving strong supervisory support [10].

Additionally, some studies among hospital nurses have suggested that burnout might be an important mediator in the links between nursing work environment and some job outcome variables such as turnover intention and ITS [13-14]. Nurses who perceived their work environment as negative experienced more burnout than those who did not [15]. Burnout reduces nurses’ ITS [14], increases nurses’ turnover intention [13]. Burnout resulting from an unfavorable work environment increased nurses’ turnover intention [13], reduced nurses’ ITS [14]. Moreover, a study has shown that nurses’ greater participation in hospital affairs reduces their likelihood of burnout [16]. Another study has found that better relationships between nurses and physicians reduce a likelihood of nurses’ burnout [14]. Other study examining the association between five aspects of nursing work environment and burnout showed that nurse burnout was significantly linked to nursing foundations for quality of care, nurse participation in hospital affairs, and adequate staffing [17]. Research has also found that when nurses are provided with sufficient support from supervisors and organizational support, they tend to experience lower levels of burnout [18,19]. Reduced burnout with adequate supervisory support may lead to improved ITS. Although many studies have investigated the factors affecting nurses’ ITS, we know little about the mechanisms underlying the relationships among five aspects of nursing work environment, supervisory support, burnout, and ITS.

Point 3: Also, the research rationale should be refined at the end of the introduction while presenting the model.

The research rationale has been refined at the end of the introduction while presenting the model.

Updated sentences are as follows.

Nurse burnout has been found to be negatively related to nurses’ ITS [9]. Additionally, some studies among hospital nurses have suggested that burnout might be an important mediator in the links between nursing work environment and some job outcome variables such as turnover intention and ITS [13-14]. Nurses who perceived their work environment as negative experienced more burnout than those who did not [15]. Burnout reduces nurses’ ITS [14], increases nurses’ turnover intention [13]. Burnout resulting from an unfavorable work environment increased nurses’ turnover intention [13], reduced nurses’ ITS [14]. Moreover, a study has shown that nurses’ greater participation in hospital affairs reduces their likelihood of burnout [16]. Another study has found that better relationships between nurses and physicians reduce a likelihood of nurses’ burnout [14]. Other study examining the association between five aspects of nursing work environment and burnout showed that nurse burnout was significantly linked to nursing foundations for quality of care, nurse participation in hospital affairs, and adequate staffing [17]. Research has also found that when nurses are provided with sufficient support from supervisors and organizational support, they tend to experience lower levels of burnout [18,19]. Reduced burnout with adequate supervisory support may lead to improved ITS. Although many studies have investigated the factors affecting nurses’ ITS, we know little about the mechanisms underlying the relationships among five aspects of nursing work environment, supervisory support, burnout, and ITS. A theoretical model can broaden our understanding of the structural relationships between ITS and its associated factors. Cowden and Cummings [9] presented a theoretical model of hospital nurses’ ITS. The model included several variables that directly or indirectly affect nurses’ ITS: work characteristics, organization characteristics, manager characteristics, and nurse characteristics. Moreover, In the model, nurses’ affective responses to work (e.g., burnout, job satisfaction) have a mediating position between work characteristics, organization characteristics, manager characteristics, and nurse characteristics, and ITS [9]. ITS is affected by nurses’ affective responses to their work environment such as burnout. Cowden and Cummings have proposed that affective responses (e.g., burnout) are not only the important predictors of ITS but also play mediating roles in ITS [9,20].

Based on Cowden and Cummings's model and other literature, we used a conceptual model to guide the present study (Figure 1). The model of this study proposes that nurses’ ITS is directly influenced by nursing work environment (work characteristics and organization characteristics), supervisory support (manager characteristics), and nurse characteristics and is indirectly influenced by these variables through burnout (affective responses). Burnout will mediate the relationships among nursing work environment, supervisory support, nurse characteristics, and ITS. In particular, we analyzed which of the five aspects of the nursing work environment contributed the most to nurses’ ITS. The current study attempts to reveal the potential mechanisms between five aspects of the nursing work environment, supervisory support, nurse characteristics, and burnout, and ITS among hospital nurses. The nursing work environment is highly associated with nurses' ITS, and more empirical evidence are needed to investigate the mechanism underlying these relationships among nurse. Our study focuses on the mediating role of burnout in the links among variables. This study aimed to examine the comprehensive impact of five aspects of the nursing work environment, supervisory support, nurse characteristics, and burnout on ITS among Korean hospital nurses.

Point 4: The conclusions should be adapted to the results of the current study and the innovation in it should be emphasized.

The conclusions section has been updated to adapt to the results of the current study. The innovation of this study has been emphasized in the conclusions.

Updated sentences are as follows.

In the conclusions section

This study firstly explored the specific causal pathways between the five aspects of nursing work environment, supervisory support, nurse characteristics, burnout, and ITS. The current study provides nurse managers with some meaningful implications to enhance ITS of nurses. In terms of nursing work environment, nurse participation in hospital affairs and collegial nurse-physician relationships showed significant direct positive effects on ITS unlike the other aspects of nursing work environment. Especially, burnout was the most influential predictor of ITS. Burnout played a mediating role on the relationships among the staffing and resource adequacy, collegial nurse-physician relationships, supervisory support, marriage, and ITS in this study. In order to increase nurse’ ITS, hospital executives and nurse managers should take corresponding measures to reduce nurse burnout and improve nursing work environment continuously.

Thank you so much.

Reviewer 3 Report

1. The references in the introduction section are too outdated. A more cutting-edge perspective is needed to present the implications of this study. For example, the article referenced in this study on burnout as a mediating variable is ten years old.

2. Since the previous theoretical model was used in this study, the role of mediating variables likewise needs to be explained in the context of theory, not just proven through existing research.

3. Validity testing is required.

4. The discussion section of this study continues to analyse each variable separately and is consistent with most previous studies. This has resulted in the innovative aspects of this study not being well highlighted. Also, the practical recommendations made should be articulated around the relationship between the variables rather than just one variable.

5. There is a lack of comparative analysis between the variables in this study. Since this study is to focus on which variable has a greater impact, a comparative analysis is needed.

6. Further directions need to be discussed in relation to the limitations of this study.

Author Response

Thank you for the thoughtful review of the ms. Entitled “Nursing work environment, supervisory support, nurse characteristics, and burnout as predictors of intent to stay among hospital nurses in South Korea: A path analysis” We have revised the manuscript following the suggestions and recommendations made in the reviews and are resubmitting it for the consideration for publication.

We have responded to the critical points in the review in the following way. The comments are italicized.

Reviewer 3

Point 1: The references in the introduction section are too outdated. A more cutting-edge perspective is needed to present the implications of this study. For example, the article referenced in this study on burnout as a mediating variable is ten years old.

The references in the introduction section have been updated with recent literature.

In the introduction section, a more cutting-edge perspective has been added to present the implications of this study.

Added sentences are as follows.

Globally, the shortage and high turnover rate of nurses are significant issues that threaten both patient safety and quality of nursing care [1]. Nurses are essential members of the workforce in the health care system. Retaining nurses who already work in hospitals would be a partial solution to the nursing shortage [2]. The intent to stay (ITS) at work has been known to be the strongest predictor of nurse retention [3]. ITS is nurses’ perception of their likelihood of staying in the current job [2]. A better understanding of nurses’ ITS may lead to improvement in nurse retention and quality of patient care. Policymakers and nurse leaders who aim to design effective nurse retention strategies must understand the variables that are associated with nurses’ ITS.

Studies of nurses in hospitals have shown a positive association between favorable work environments and ITS [4,5]. Nurses working in better work environments have reported higher ITS. The nursing work environment is defined as the organizational characteristics of a work setting that foster or hamper professional nursing practice [6]. Lake [6] suggested that the nursing work environment is composed of five aspects: nurse participation in hospital affairs; nursing foundations for quality care; nurse managers’ ability, leadership, and support; staffing and resource adequacy; and collegial nurse-physician relationships. Although numerous studies have investigated the association between nursing work environment and ITS, we know little about which of these five aspects contributes the most to nurses’ ITS. To our knowledge, no studies have examined the link between these five aspects of nursing work environment and ITS among hospital nurses. Knowledge about how various aspects of the nursing work environment relate to ITS is needed to develop optimal policies for enhancing nurses’ ITS.

Supervisory support is defined as the assistance and caring that employee receive from supervisors or managers [7]. In a hospital setting, supportive supervisor can help nurses do their jobs more effectively and provide work guidelines to improve work processes [8]. Manager characteristics such as supervisory support, praise, recognition, and leadership have also been known to affect nurses’ ITS [1,9]. Nurses showed an increased ITS when perceiving strong supervisory support [10]. Nurses’ personal characteristics, such as age, experience as a nurse, marriage, education, and position could also influence ITS [9,11,12]. Older nurses were more likely to remain employed than younger nurses [11]. The intent to stay increased as the nurses’ years of experience increased [12]. Married, and less educated nurses are more likely to remain in their current positions than their counterparts [1,9].

Nurse burnout has been found to be negatively related to nurses’ ITS [9]. Additionally, some studies among hospital nurses have suggested that burnout might be an important mediator in the links between nursing work environment and some job outcome variables such as turnover intention and ITS [13-14]. Nurses who perceived their work environment as negative experienced more burnout than those who did not [15]. Burnout reduces nurses’ ITS [14], increases nurses’ turnover intention [13]. Burnout resulting from an unfavorable work environment increased nurses’ turnover intention [13], reduced nurses’ ITS [14]. Moreover, a study has shown that nurses’ greater participation in hospital affairs reduces their likelihood of burnout [16]. Another study has found that better relationships between nurses and physicians reduce a likelihood of nurses’ burnout [14]. Other study examining the association between five aspects of nursing work environment and burnout showed that nurse burnout was significantly linked to nursing foundations for quality of care, nurse participation in hospital affairs, and adequate staffing [17]. Research has also found that when nurses are provided with sufficient support from supervisors and organizational support, they tend to experience lower levels of burnout [18,19]. Reduced burnout with adequate supervisory support may lead to improved ITS. Although many studies have investigated the factors affecting nurses’ ITS, we know little about the mechanisms underlying the relationships among five aspects of nursing work environment, supervisory support, burnout, and ITS. A theoretical model can broaden our understanding of the structural relationships between ITS and its associated factors. Cowden and Cummings [9] presented a theoretical model of hospital nurses’ ITS. The model included several variables that directly or indirectly affect nurses’ ITS: work characteristics, organization characteristics, manager characteristics, and nurse characteristics. Moreover, In the model, nurses’ affective responses to work (e.g., burnout, job satisfaction) have a mediating position between work characteristics, organization characteristics, manager characteristics, and nurse characteristics, and ITS [9]. ITS is affected by nurses’ affective responses to their work environment such as burnout. Cowden and Cummings have proposed that affective responses (e.g., burnout) are not only the important predictors of ITS but also play mediating roles in ITS [9,20].

Based on Cowden and Cummings's model and other literature, we used a conceptual model to guide the present study (Figure 1). The model of this study proposes that nurses’ ITS is directly influenced by nursing work environment (work characteristics and organization characteristics), supervisory support (manager characteristics), and nurse characteristics and is indirectly influenced by these variables through burnout (affective responses). Burnout will mediate the relationships among nursing work environment, supervisory support, nurse characteristics, and ITS. In particular, we analyzed which of the five aspects of the nursing work environment contributed the most to nurses’ ITS. The current study attempts to reveal the potential mechanisms between five aspects of the nursing work environment, supervisory support, nurse characteristics, and burnout, and ITS among hospital nurses. The nursing work environment is highly associated with nurses' ITS, and more empirical evidence are needed to investigate the mechanism underlying these relationships among nurses. Our study focuses on the mediating role of burnout in the links among variables. This study aimed to examine the comprehensive impact of five aspects of the nursing work environment, supervisory support, nurse characteristics, and burnout on ITS among Korean hospital nurses.

Point 2: Since the previous theoretical model was used in this study, the role of mediating variables likewise needs to be explained in the context of theory, not just proven through existing research.

In the introduction section, the role of mediating variables (e.g., burnout) has been updated to be explained in the context of theory.

Updated sentences are as follows.

In the introduction section

Nurse burnout has been found to be negatively related to nurses’ ITS [9]. Additionally, some studies among hospital nurses have suggested that burnout might be an important mediator in the links between nursing work environment and some job outcome variables such as turnover intention and ITS [13-14]. Nurses who perceived their work environment as negative experienced more burnout than those who did not [15]. Burnout reduces nurses’ ITS [14], increases nurses’ turnover intention [13]. Burnout resulting from an unfavorable work environment increased nurses’ turnover intention [13], reduced nurses’ ITS [14]. Moreover, a study has shown that nurses’ greater participation in hospital affairs reduces their likelihood of burnout [16]. Another study has found that better relationships between nurses and physicians reduce a likelihood of nurses’ burnout [14]. Other study examining the association between five aspects of nursing work environment and burnout showed that nurse burnout was significantly linked to nursing foundations for quality of care, nurse participation in hospital affairs, and adequate staffing [17]. Research has also found that when nurses are provided with sufficient support from supervisors and organizational support, they tend to experience lower levels of burnout [18,19]. Reduced burnout with adequate supervisory support may lead to improved ITS. Although many studies have investigated the factors affecting nurses’ ITS, we know little about the mechanisms underlying the relationships among five aspects of nursing work environment, supervisory support, burnout, and ITS. A theoretical model can broaden our understanding of the structural relationships between ITS and its associated factors. Cowden and Cummings [9] presented a theoretical model of hospital nurses’ ITS. The model included several variables that directly or indirectly affect nurses’ ITS: work characteristics, organization characteristics, manager characteristics, and nurse characteristics. Moreover, In the model, nurses’ affective responses to work (e.g., burnout, job satisfaction) have a mediating position between work characteristics, organization characteristics, manager characteristics, and nurse characteristics, and ITS [9]. ITS is affected by nurses’ affective responses to their work environment such as burnout. Cowden and Cummings have proposed that affective responses (e.g., burnout) are not only the important predictors of ITS but also play mediating roles in ITS [9,20].

Point 3: Validity testing is required.

Validity testing has been updated.

Updated sentences are as follows.

In the statistical analysis section

In this analysis, the following fit indices were used to evaluate the validity and fitness of the model: non‐significant χ2 value, degrees of freedom (df), Tucker-Lewis index (TLI), comparative fit index (CFI), root mean square error of approximation (RMSEA), and standardized root mean square residual (SRMR). A model is evaluated to have a good fit if the CFI and TLI are 0.95 or greater, the RMSEA value is lower than 0.08 and the SRMR value is <0.08 [31]. All study variables in our model have been rigorously tested with diverse study populations as well as in this study regarding validity and reliability [6,22-30]. In addition, we computed the squared multiple correlations (SMC) of the study variables on burnout and ITS in the path model.

In the results section

3.3. Evaluation of the path model

To examine our hypothesized path model validity, we used the non‐significant χ2 value, df, TLI, CFI, RMSEA, and SRMR. The fit indices of final model showed favorable convergent validity with the TLI = 1.000, CFI = 1.000, RMSEA = 0.000 and SRMR = 0.000. Although the chi-square value was significant (χ2 = 499.860, df = 21, p < .001), it is not appropriate for fit index when the sample size is large. The reliability of the study variables was verified via Cronbach’s alpha coefficients. As shown in Table 2, the Cronbach’s alpha values for variables were between 0.69 and 0.92.

Point 4: The discussion section of this study continues to analyse each variable separately and is consistent with most previous studies. This has resulted in the innovative aspects of this study not being well highlighted. Also, the practical recommendations made should be articulated around the relationship between the variables rather than just one variable.

The discussion section has been updated. The innovative aspects of this study have been updated to be well highlighted. The practical recommendations around the relationship between the variables have been added in the discussion section.

Added sentences are as follows.

Two of the five aspects of the nursing work environment had a direct effect on ITS. Nurse participation in hospital affairs directly influenced ITS, supporting our hypothesized model. Nurses were more likely to report higher levels of ITS when they perceive high levels of nurse participation in hospital affairs. Involving nurses in hospital activities and decisions may enhance their autonomy and job engagement, which is a crucial work-related factor that has been linked to ITS [32,33]. Our finding was the same as that found in previous studies: that lower nurse participation in hospital affairs was associated with higher intent to leave [34]. Nurse participation in hospital affairs was not significantly linked to burnout, inconsistent with previous studies [17,35]. This finding did not support the proposed hypotheses. Studies have found that nurses’ greater participation in hospital affairs reduced their likelihood of burnout [17,35]. If nurses take more active roles in decision-making for hospital affairs, they might perceive themselves as empowered and decrease their risk of burnout [17]. Further research testing clearly the association between nurse participation in hospital affairs and burnout is needed.

 Collegial nurse-physician relationships directly influenced ITS. Nurses who perceived high levels of collegial nurse-physician relationships would have higher levels of ITS. This is resembling findings of previous studies that good relationships with physicians were positively related with ITS [36]. Collegial nurse-physician relationships can improve work commitment and job satisfaction, which impact ITS [36,37]. Collegial nurse-physician relationships were also tied to lower levels of burnout, consistent with existing research [35]. These findings supported our hypotheses that collegial nurse-physician relationships are associated with ITS directly and indirectly, and burnout plays a mediating role on the relationships between collegial nurse-physician relationships and ITS. This was in line with studies that revealed collegial nurse-physician relationships were associated with lower levels of burnout [35], and burnout mediated the relationship between nurse-physician relations and turnover intentions [38].

Staffing and resource adequacy did not directly influence nurses’ ITS, but indirectly influenced ITS through the burnout. This finding partially supports our hypothesized model. Staffing and resource adequacy had a strong negative influence on burnout. Burnout held a mediating position between staffing and resource adequacy and ITS. These results support previous evidence that sufficient staffing and resources could reduce the risk of nurse burnout, thereby promoting ITS [17,35]. Chronic understaffing and resource inadequacy may cause unsustainable workloads, eventually leading to increased burnout [35]. Accordingly, in order to prevent nurses from experiencing burnout, it is crucial to keep staffing aligned with the actual job demands and retain adequate resources. unsustainable

Supervisory support directly influenced ITS. Nurses who perceived high levels of supervisory support would have higher levels of ITS. This is in accord with prior literature reporting the associations between supervisory support and ITS, turnover intention in nurses [9,39]. Supervisory support can make nurses feel that their work is appreciated, to be satisfied with their job, and encourage them to intend to stay at work [11]. Supervisory support also had a negative influence on burnout, in line with prior investigations [40]. Nurses who experience sufficient support from supervisors appear better able to cope with the potential negative effects of work pressures and can experience less burnout [40], more job satisfaction [39], leading to enhance ITS [31]. These findings supported our hypotheses that supervisory support is link to ITS directly and indirectly, and burnout plays a mediating role on the relationships between supervisory support and ITS. Our finding demonstrates the importance of supervisory support in reducing nurse burnout, further enhancing ITS. In the present study, only two nurse characteristics (age and education) were found to have significant direct effects on nurses’ ITS. Older nurses were more likely to stay with their job than younger nurses, consistent with previous findings [1]. Nurses with bachelor’s degrees and above were more likely to stay with their job than nurses with a diploma, which is dissimilar to other findings [25]. Marriage had a negative direct effect on nurse burnout. Married nurses had lower burnout compared with unmarried nurses, inconsistent with existing findings that married healthcare workers have higher burnout level than their unmarried counterparts because of family roles and work conditions [41].

In the current study, burnout directly and negatively affected ITS, supporting our hypothesized model. When nurses are burned out, they may feel they are not doing their best to care for patients, cannot communicate properly with others, lack motivation, and have poor job-related self-esteem [42]. Eventually, burnout may contribute to nurses leaving their positions [42]. Burnout also played a mediating role on the relationships among the staffing and resource adequacy, collegial nurse-physician relationships, supervisory support, marriage, and ITS in this study. In addition, burnout was the most crucial predictor of ITS in the present study. This finding highlights the importance of burnout for enhancing ITS of nurses. Thus, nurse managers should systematically measure burnout and design proper tactics to reduce burnout of nurses. Furthermore, we examined the effect of each aspect of the nursing work environment on ITS. The findings showed that among the five aspects, nurse participation in hospital affairs had direct positive effect on ITS and collegial nurse-physician relationships had direct and indirect positive effect on ITS. To our knowledge, the present study was the first study to explore how five aspects of the nursing work environment, supervisory support, nurse characteristics, and burnout contribute to ITS among hospital nurses. Our findings extend understanding of how five aspects of the nursing work environment, supervisory support, and nurse characteristics are related to ITS through burnout. These results are important, because it provides nurse managers with empirical guidance to help to improve ITS of nurses.

Point 5: There is a lack of comparative analysis between the variables in this study. Since this study is to focus on which variable has a greater impact, a comparative analysis is needed.

Comparative analysis between the variables has been undertaken. The results have been discussed in the discussion section.

Added sentences are as follows.

In the results section

Additionally, in order to determine which variable has the greater effect on ITS, a comparative analysis of the standardized total effects (β) of the variables on ITS was undertaken. The variables that had the greater impact on ITS were burnout (β = −0.36, p < .001), age (β = 0.28, p < .001), supervisory support (β = 0.24, p < .001), nurse participation in hospital affairs (β = 0.14, p = .045), and collegial nurse-physician relationships (β = 0.14, p = .039), in that order.

In the discussion section

In addition, burnout was the most crucial predictor of ITS in the present study. This finding highlights the importance of burnout for enhancing ITS of nurses. Thus, nurse managers should systematically measure burnout and design proper tactics to reduce burnout of nurses. Furthermore, we examined the effect of each aspect of the nursing work environment on ITS. The findings showed that among the five aspects, nurse participation in hospital affairs had direct positive effect on ITS and collegial nurse-physician relationships had direct and indirect positive effect on ITS. To our knowledge, the present study was the first study to explore how five aspects of the nursing work environment, supervisory support, nurse characteristics, and burnout contribute to ITS among hospital nurses. Our findings extend understanding of how five aspects of the nursing work environment, supervisory support, and nurse characteristics are related to ITS through burnout. These results are important, because it provides nurse managers with empirical guidance to help to improve ITS of nurses.

Point 6: Further directions need to be discussed in relation to the limitations of this study.

Further directions have been discussed in relation to the limitations of this study.

Added sentences are as follows.

It is vital to note limitations of this research, and directions for further studies. This study had several limitations. First, we used cross-sectional data, which may have limited causality. Thus, a longitudinal study is warranted to confirm causal relationships between the nursing work environment, supervisory support, nurse characteristics, burnout and ITS. Second, our measures of the nursing work environment were self-reported and could not determine actual work environment characteristics. Third, since we used convenience sampling in this study, the generalization of the findings might be limited. The revised model resulting from this study needs to be tested in future studies with a more diverse cohort. Further research using a randomized sampling technique with longitudinal designs is warranted to generalize and extend the study findings. Despite these limitations, our study adds important information to the current literature by exploring the comprehensive impact of five aspects of the nursing work environment, supervisory support, nurse characteristics, and burnout on ITS among Korean hospital nurses.

Thank you so much.
